# Multilevel Toxicity Evaluations of Polyethylene Microplastics in Zebrafish (*Danio rerio*)

**DOI:** 10.3390/ijerph20043617

**Published:** 2023-02-17

**Authors:** Ingrid de Souza Freire, Maria Luiza Fascineli, Tathyana Benetis Piau, Bruno Fiorelini Pereira, Cesar Koppe Grisolia

**Affiliations:** 1Departamento de Genética e Morfologia, Instituto de Ciências Biológicas, Universidade de Brasília, Brasília 70910-900, Brazil; 2Departmento de Ciências Biológicas, Universidade Federal de São Paulo—UNIFESP, São Paulo 05468-901, Brazil

**Keywords:** microplastics, ecotoxicology, zebrafish, polyethylene, biomarkers

## Abstract

Microplastics in freshwater environments pose a serious threat to living beings. Polyethylene microplastics (PE-MP) are the type most used around the world as microbeads in personal care products, and they have been found in aquatic organisms. The behavior and toxicity of fluorescent PE-MP spheres with an average diameter of 58.9 μm were studied in adult, juvenile and embryo zebrafish (*Danio rerio*). The adults were studied for genotoxicity, cytotoxicity, histology and biochemical markers. Juveniles underwent a follow-up in the gastrointestinal (GI) tract with histologic observations, and embryos were studied for embryotoxicity with the FET-test. In adults, micronucleus test and comet assays found neither genotoxicity after acute exposure for 96 h at concentrations of 0.0, 12.5, 50 and 100 mg.L^−1^, nor cytotoxicity through the nuclear abnormalities test. Acetylcholinesterase (AChE), Glutathione-S-Transferase (GST) and Lactate Dehydrogenase (LDH) activities were measured in adults exposed for 96 h. The AChE and GST activities were significantly changed, while no changes occurred for LDH. In conclusion, these PE-MP spheres did not cause serious toxic effects in zebrafish because there was no internalization. The observed biochemical changes in AChE and GST may be associated with GI microbiological dysbiosis, previously reported. The PE-MP spheres in the intestine of juveniles remained present for 12–15 days on average after the post-exposure clearance study, showing a slow depuration. The histological analysis, in adults, found no internalization of these microbeads, with complete depuration. The PE-MP spheres did not cross the chorion barrier, showing no embryotoxic effects after exposures at 0.0, 6.25, 12.5, 50.0 or 100.0 mg.L^−1^ for 96 h.

## 1. Introduction

Plastics in the environment are degraded, producing fragments, microplastics and nanoplastics, which have been found at different trophic levels. They are considered major emerging pollutants around the world, and their toxicities are not completely understood [1]. In the process of fragmentation, a variety of shapes and sizes have been found, such as pellets, fibers, films and spheres [2,3]. Microplastics (MPs) have been found in almost all marine organisms, and due to the huge concern regarding their end point and effect on marine biota, many studies have been published [4]. On the other hand, the ecotoxicity of microplastics in freshwater environments should also be further investigated, because this water is used as a drinking water resource [5]. MPs in freshwater have not yet been investigated as much as in oceans. In both cases, MPs enter the food chains, reaching top-chain organisms. The differences in size and shape of MPs influence their uptake and accumulation in tissues in many animal species, as already found in zebrafish. Due to their poor degradation and persistence, MPs remain available in aquatic environments for a long time [6]. 

Plastic pellets are manufactured as microbeads for use in cosmetic and personal care products. Over 90% of microbeads used in cosmetics consist of spherical particles of polyethylene with size varying from 0.4 nm to 1.24 mm [7]. Most microbeads have been found in waters near large cities, especially in developing countries with limited waste treatment systems [8]. European countries and many other countries around the world have banned plastic microbeads in personal care products, and in Brazil, some states have followed a federal law on solid residues, which they use to prohibit the use of non-biodegradable plastic bags in local stores [9]. Although these measures have been adopted, significant levels of polyethylene microbeads still remain in the environment due to their low degradation [10]. In Brazil, among plastic polymers, low- and high-density polyethylenes are the type most produced by chemical industries [11]. 

In Brazil, studies have revealed the presence of polyethylene in coastal sediments [12] and on the seacoast of several states [13,14]. Another study states that polyethylene plastics are also widely found in the digestive tract of different aquatic or terrestrial species, such as fish, amphibians and microcrustaceans. As for the size of these particles, there is great diversity, and they are also present in freshwater, seawater and estuary environments [15]. Plastic thus achieves entry into different levels of the food chain, becoming one of the most worrying pollutants today [16].

The zebrafish in vivo model was used in this study due to its high similarity with mammals, including human beings. Zebrafish have been used for ecotoxicological studies with numerous emerging pollutants, such as endocrine disruptors, pharmaceutics, pesticides and nanomaterials. In addition, zebrafish share high homology with the human genome, having approximately 70% orthologue human genes [17,18]. The purpose of this study was not only to investigate the toxicity and genotoxicity of polyethylene MP beads in zebrafish, but also to examine their interaction and behavior when ingested. 

## 2. Materials and Methods

### 2.1. Polyethylene Microplastics (PE-MP)

Fluorescent orange PE-MPs with diameter of 53–73 μm on average were acquired from Cospheric (Goleta, CA, USA). The PE-MPs were documented using scanning electron microscopy (SEM—Jeol JSM-7001F at 15 kV, Tokyo, Japan). The samples were previously coated with gold powder for their analysis and imaging performed using an electron microscope. The composition of the PE-MPs was analyzed using energy-dispersive X-ray spectroscopy (EDS). Using the software FijiJ, 500 PE-MPs were scored, exhibiting 58.9 ± 4.52 μm average diameter (Figure 1A,B). Orange fluorescence was demonstrated under a Axioskop 2—Zeiss (Jena, Germany) epifluorescence microscope with magnification of 100× using a 510–530 nm filter (Figure 1C). In the pilot test, this MP agglomerated in water, after which Tween 80% was used in all tests as a dispersant.

### 2.2. Zebrafish (Danio rerio) as Test Organism

Zebrafish adults from a University of Brasilia facility (ZebTec—Tecniplast, Varese, Italy) were raised under controlled water parameters: temperature at 27.0 ± 1 °C, conductivity at 650 ± 100 μS/cm, pH at 7.0 ± 0.5 and dissolved oxygen ≥ 95% saturation. Zebrafish embryos were obtained using the Ispawn breeding system (Tecniplasty, Italy). 

### 2.3. Fish Embryo Toxicity Test (FET)

The FET test followed the OECD guideline Protocol 236 [19]. The embryos were exposed to 6.25, 12.5, 50.0 and 100.0 mg.L^−1^ of PE-MPs and Tween 80% at 0.01% for 96 h; there was also a control group. The embryos were collected using the Ispawn system after natural mating. Unfertilized eggs and those with cleavage irregularities or injuries were discarded. The test was performed using 60 embryos per treatment in 3 replicates, randomly selected and distributed in 24-well microplates in the climate chamber. Embryos and larvae were observed daily under a reverse microscope. The following parameters were evaluated: coagulation, otolith delay in development, eye and tail pigmentation, somite formation, heartbeat frequency alterations, edemas, detachment of the tail from the yolk sac, yolk sac absorption and hatching. Body malformations were evaluated and documented as recommended by the OECD protocol. In addition, larvae were observed for up to 7 days after fertilization to check whether their feeding habits would lead to ingestion of the particles. 

### 2.4. Juveniles Test

To verify how long it took for particles to depurate, 40-day-old juveniles (n = 8) [20] were exposed for 72 h to 50 mg.L^−1^ of PMs. At the end of the exposure time, the water was renewed without microplastics (recovery), and the juveniles were monitored over the next days under a fluorescence microscope to check for the presence or absence of microparticles in the gastrointestinal tract. 

### 2.5. Adult Toxicity Test: Histology, Micronucleus, Comet Assay and Biochemical Biomarkers

Eight-month-old zebrafish were exposed in groups of seven fish to MPs at concentrations of 12.5, 50 and 100 mg.L^−1^, as well as Tween 80% (0.01%) and water (control) for 96 h for micronucleus, nuclear abnormalities, comet assay, histology and biochemical biomarkers. Fish were randomly chosen for each studied endpoint and then anesthetized using tricaine (5%), followed by spinal disruption. The tests were based on a modified version of OECD guideline Protocol n. 203 [21] and were carried out in triplicate. Tween 80% at 0.01% was used due to the hydrophobicity of polyethylene MPs [22,23]. In this study, zebrafish adults were exposed to MPs at concentrations of 12.5, 50 and 100 mg.L^−1^ for 96 h. 

#### 2.5.1. Histology

The fish were stored in Davidson fixative for 24 h and kept in 15 mL tubes with isopropyl alcohol 70%. Decalcification was performed with EDTA 4% for 7 days with total renewal of the solution after 4 days, following the literature [24]. Next, the samples were washed with ultrapure water and dehydrated in an increasing concentration gradient of isopropyl alcohol (70% for 30 min, 95% three times for 15 min, 100% three times for one hour). The fish were embedded in paraffin blocks. The blocks were chilled at 4 °C and cut with 5 μm thickness. To remove the rest of the paraffin, the slides were placed in an oven at 65–70 °C and then washed in isopropyl alcohol 100% until the paraffin was removed, as proposed in the literature [25].

#### 2.5.2. Genotoxicity and Cytotoxicity 

For micronucleus, nuclear abnormalities and comet assay, peripheral blood samples were homogenized in 200 µL of fetal bovine calf. From this sample, 50 µL were used for smear in the micronucleus (MN) and nuclear abnormalities (NA) study following the literature [26,27]: 2000 erythrocytes were scored for MN and 3000 for NA at 1000× magnification, and they were evaluated under a blind code. Erythrocytes were also scored to classify nuclear abnormalities such as BB—blebbed, LB—lobed, NT—notched, BN—binucleated and NB—nuclear bud. For comet assay, exposures occurred at 12.5, 50, 100 mg.L^−1^, Tween 80% (0.01%) and water control for 96 h. In the alkaline test, 20 µL of the blood sample was homogenized in 100 µL of LM agarose at 0.5% at 37 °C, then these samples were distributed on microscope slides, matte and polished tip, and covered with a coverslip of 60 mm. The slides were kept in lyse solution for 1 h at 4 °C, and electrophoresis occurred at 0.85 V/cm and 4 °C for 15 min. The cell (nucleoid) analysis for comet classifications followed the protocol developed by Sing et al. (1988) [28], with modifications. For positive control, H_2_O_2_ at 0.1% was used. One hundred nucleoids per fish were analyzed (blind analysis) and classified based on tail length. The damage index was measured as moderate damage (% MD), comets classified at levels 1 and 2, and high damage (% HD), comets levels 3 and 4 [29].

#### 2.5.3. Biochemical Biomarkers 

For acetylcholinesterase (AChE) measurement, samples from head and tail were used. The head was considered from mouth until operculum and tail were considered from anal pore until caudal fin tip. For the measurement of glutathione-S-transferase (GST) activity, gill and body samples were used. We took body samples, considering from operculum until anal pore. For LDH, only samples from the tail were used. The samples were stored in phosphate buffer 0.1 M, pH 7.2, at −20 °C. Samples were defrosted on ice, triturated using scissors, homogenized using a sonicator, and centrifuged while refrigerated (4 °C) for 20 min at 10,000 g. The resulting post-mitochondrial supernatant (PMS) was isolated and placed in 96-well microplates for enzymatic determinations performed spectrophotometrically (Thermo, Waltham, MA, USA) in quadruplicate. 

The Bradford method was used to quantify proteins [30]. The reactions were performed spectrophotometrically in quadruplicate, with the protocol adapted for microplates [31].

AChE activity was carried out using acetylthiocholine (ASCh) and propionylthiocoline (PSCh) as substrates. We followed the protocol described by Ellman and coworkers [32], with modifications (absorbance at 414 nm, every 40 s, for 5 min). AChE was expressed as nanomol of substrate hydrolyzed per minute and per mg of protein (U) after 10 min of absorbance reaction for measurement of enzymatic activity. Lactate Dehydrogenase (LDH) activity was performed using pyruvate as substrate and measuring the reduction of pyruvate and the oxidation of NADH at 340 nm, every 40 s, for 5 min. The determinations of the LDH activity followed the protocol described by Vassault [33]. The variations in Glutathione-S-Transferase (GST) activity were carried out according to the method of Habig and coworkers [34].

### 2.6. Statistical Analysis

The software SigmaPlot version 12.5 was used for ANOVA one-way testing, after the Shapiro–Wilk normality test was performed. The Kruskal–Wallis test was used for non-parametric analysis. The Dunnett test was used to compare differences among exposure concentrations with controls for both parametric and non-parametric tests at a level of 5%.

## 3. Results

### 3.1. Fish Embryo Toxicity Test (FET)

The polyethylene microplastics did not cause any alterations in zebrafish embryogenesis. There was no hatching delay, induction of malformations or increased mortality (Figure 2, ANOVA *p* > 0.05). Therefore, microbeads with average diameter of 58 μm do not cross the chorionic membrane, sticking to the outer side of the chorion (Figure 3A). In Figure 3B, the zebrafish larva gut is completely full of microbeads, showing that the contamination starts only after hatching. 

### 3.2. Juveniles Test

After 15 days in clean water, post-exposure period, the agglomerate of MPs disappeared from the lumen, showing that the gastrointestinal (GI) tract slowly eliminated those MPs (recovery test, Figure 4).

### 3.3. Histology 

The histological analysis shows that PE-MPs strongly agglomerated with fecal content in the intestinal lumen (Figure 5B,E,F). Due to their fluorescence, they were clearly detectable in the lumen (Figure 5E), but not in the intestinal wall or villi. MP particles were not seen in the gill and liver (Figure 5C,D,F).

### 3.4. Genotoxicity and Cytotoxicity

The MN-test did not evidence chromosome breaks or chromosome malsegregation, and nor were nuclear abnormalities seen for all exposure levels tested (Figure 6A, *p* > 0.05). In the comet test, with the tail length in the nucleoids analyzed for all exposure levels, no increased DNA break indices were observed (Figure 6B, *p* > 0.05), except for the positive control H_2_O_2_ at 0.1%, *p* < 0.05.

### 3.5. Biochemical Biomarkers

The measurement of AChE activities in the zebrafish adult head demonstrated significant differences at the exposure levels of 50 and 100 mg.L^−1^ compared with the control group (*** *p* < 0.001). The results in tail measurement did not show a difference in activity compared with the control (*p* > 0.05), as seen in Figure 7A. The polyethylene microplastics did not interfere with this neuronal transmitter. The LDH activity in the tail was not modified when compared with the control group (Figure 7B, *p* > 0.05).

The GST activity tests evidence a greater variation due to polyethylene microplastics exposures when compared among the other biochemical markers studied. In the body, a significant decrease was observed in GST activity at 50 and 100 mg.L^−1^ (* *p* < 0.05). On the other hand, the activity measured in gills increased and demonstrated a concentration–effect relationship (*** *p* < 0.001), shown in Figure 7C.

## 4. Discussion

Industrial and domestic wastewater contains fragments, pellets and fibers of plastics that are not removed by wastewater treatment plants due to their small size and buoyancy [7]. Microplastics have been found around the world in soil, water and air, and sooner or later they end up in the human body. In an Indonesian population study, MPs were detected in the gastrointestinal tract, with an average of 10 µg/g of feces [35]. Airborne MPs, commonly found in many cities, are inhaled and their presence in human lung tissues has already been detected in autopsies [36]. Human placentas collected from physiological pregnancies and analyzed using Raman spectroscopy have shown the presence of MPs with different shapes [37]. Due to the fact that different types of MPs have already been reported in humans, more studies to understand the fate and toxicity of microplastics within the human body must be performed.

The zebrafish embryotoxicity test is recommended to assess the adverse potential of emerging aquatic pollutants, such as plastic, and this test aims to determine the toxicity of substances in the embryonic stages of the fish, evaluating the formation of the embryo or larvae during the development [19,38]. In this study, no embryotoxicity occurred from these MPs due to the protection of the chorion, acting as an effective barrier (Figure 3A, *p* > 0.05). The zebrafish chorion is a membrane with thickness of 3.5 µm with 0.5–0.7 µm diameter pores, allowing the passage of water, ions and some chemicals [39]. The chorion acts as a barrier, preventing compounds larger than 3 kDa from passing freely until the embryo hatches at around 72 hpf. When the mouth starts to open at 60 hpf, oral uptake gains importance as a route of exposure [40]. However, after hatching, larvae start feeding freely and are contaminated, as demonstrated in Figure 3B. This result contradicts the studies performed by Malafaia and coworkers [38], in which polyethylene particles with red fluorescence and an average diameter of 38.26 µm, smaller than those used in this study, provoked early hatching of embryos and altered their morphometric standards in static exposure. In contrast, another study [41] found that a polystyrene nanoplastic of 22 nm did not induce significant morphological alterations in zebrafish embryos, as was observed in our study.

In adults, these MPs were easily ingested and were found throughout the GI tract (Figure 4). Their gastrointestinal tract is similar to that of other vertebrates. All cyprinids lack a stomach, and the intestine is continuous, composed of stratified squamous epithelium [42]. Polystyrene MPs of 0.5 and 50 μm diameter caused microbiota dysbiosis and inflammation in the intestines of adult zebrafish [43]. Additionally, Jin et. Al. [44], studying MPs of 50 µm in zebrafish exposed for 14 days, did not observe histopathological changes in the intestine, but instead a significant change in the microbiota diversity leading to dysbiosis. Oxidative stress and cytotoxicity as a consequence of inflammatory processes have been found in zebrafish exposed to MPs. They can be absorbed by macrophages, contributing to enhancing reactive oxygen species and altering metabolic enzymes [45].

Microplastics of different sizes and shapes have been found in fish intestines around the world, causing several physiological disturbances. They can be accumulated and translocated within an organism or excreted through feces, depending on the size, shape or level of exposure. Spherical MPs (microbeads) cause less injury to the intestine and are retained for a shorter time than those with irregular shapes [46,47]. In our study, the spherical PE-MPs of 58 µm at 50 mg.L^−1^ for 96 h demonstrated an intestinal transit of 12–15 days on average, and we did not find these MPs in other zebrafish tissues. This supports other studies suggesting that the rapid removal of plastic from aquatic organisms is a strategy taken by these species to reduce potential biological effects [8].

Analyses of DNA damage have been efficient in detecting genotoxic environmental contaminants, even in low concentrations [48]. Araújo et al. [49] verified that polyethylene microplastics with a diameter of 36 µm induced DNA breaks in zebrafish after exposure of 15 days, using comet assay. However, their results from the micronucleus test and nuclear abnormalities were not statistically significant. The comet assay is a more sensitive endpoint than the micronucleus test. Considering the biochemical markers used, it is known that the assessment of AChE levels has been used as an indicator of in vivo neurotoxicity and the GST is involved with detoxification xenobiotics [50]. In the present study, MPs caused alterations in some biochemical biomarkers, such as AChE inhibition in the zebrafish head and GST increased activity in the gills. AChE activity measurement is a biomarker of neurologic adverse effects, and in this case, this could have occurred due to circulating pro-inflammatory cytokines, commonly seem in intestinal dysbiosis [51].

The GST enzymes exhibit different patterns of expression in tissues, and their main functions are the detoxification of xenobiotics. Environmental toxicants in water reach the gill first, and their higher activity in this tissue therefore makes sense, even being expressed also in other tissues of the fish body. The GST metabolic enzymes were already well-characterized in zebrafish by Glisica et al. [52], showing a comprehensive study of this gene family. The authors showed the involvement of seven different GST classes in adult zebrafish associated with the biotransformation of xenobiotics, as well as their functional similarity to the human orthologue xenobiotic metabolism. Taking into account that these microbeads remain in water, our results clearly showed their effects mainly on gill GST metabolism in adult zebrafish.

Lactate dehydrogenase (LDH) activity is commonly used to measure cytotoxicity caused by xenobiotics, leading to the disruption or lysis of cell membranes. When tissues are damaged, LDH is released in response to cell injuries [53]. Our results did not show increases in the LDH levels, from which it can be understood that there was no cell injury. In fact, this could be a survival strategy that fish have developed to eliminate pollutants quickly, in order to avoid other biological side-effects [8]. Our results clearly show the presence of PMs in the lumen of the intestine (Figure 5E) but not in the other organs, it means that there was not absorption (Figure 5C,D). The Figure 5B clearly shows MPs completely agglomerated, like pellets of MPs, which may make it difficult for the intestinal mucosa to absorb.

In a review article, Prüstt et al. [54] affirmed that micro- and nanoplastics can inhibit the activity of AChE and increase GST. Araújo et al. [49] also confirmed alterations in biochemical stress markers, such as superoxide dismutase (SOD) and catalase (CAT), in zebrafish after exposure of 15 days to polyethylene microplastics. Another work [41] found biochemical alterations in the markers AChE, GST and CAT in zebrafish exposed to polystyrene nanoplastics, with a diameter of 22 nm, in acute exposure, which corroborates our study.

## 5. Conclusions

Spherical MPs (microbeads) caused physiological disturbances in zebrafish after short-time exposures, mainly with AChE and GST activity changes. The alterations in GST activity may be associated with oxidative stress caused by a possible intestinal dysbiosis, which was not effective enough to induce genotoxicity and cytotoxicity. We also demonstrated that the chorion protects embryos against MPs of 58 µm, and the microparticle presents complete intestinal transit and is not absorbed, as it was not found in other tissues. Even with the high concentrations seen in the intestine, it was confirmed that the zebrafish manages to eliminate this type of MP.

## Figures and Tables

**Figure 1 ijerph-20-03617-f001:**
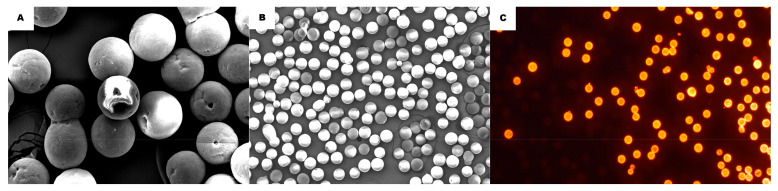
(**A**,**B**) images from SEM showing the morphology of the PE-MPs (magnification (**A**)—500×, (**B**)—370×). (**C**) shows the fluorescence of the PE-MPs in shape of beads (magnification 100×).

**Figure 2 ijerph-20-03617-f002:**
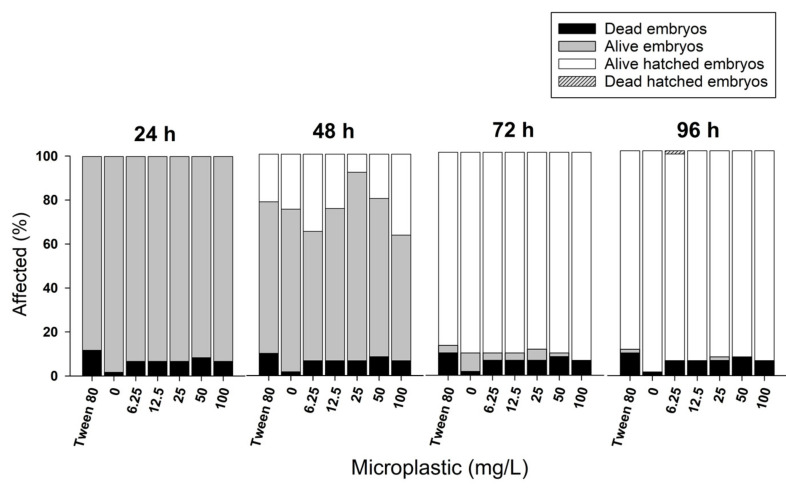
Results of the embryo toxicity test after 96 h of exposure, showing no alterations in the analyzed parameters compared with controls (ANOVA, *p* > 0.05).

**Figure 3 ijerph-20-03617-f003:**
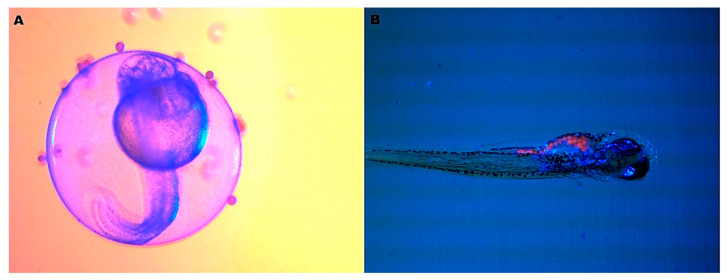
(**A**) Zebrafish embryo 36 hpf exposed at 50 mg.L^−1^, showing plastic microbeads adhered on the outside of the chorionic membrane. (**B**) Microbeads shown in the gut of 7-day-old larva ((**A**)—25× and (**B**)—15× magnification, stereomicroscopy image).

**Figure 4 ijerph-20-03617-f004:**
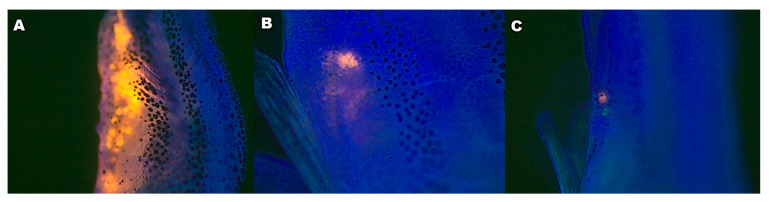
(**A**) Juveniles exposed to 50 mg.L^−1^ for 24 h, deposition of particles in the gastrointestinal tract. (**B**) After 7 days of recovery, exposed only in water for depuration. (**C**) After 12 days, few particles are observed in the anal pore. Magnification 40×.

**Figure 5 ijerph-20-03617-f005:**
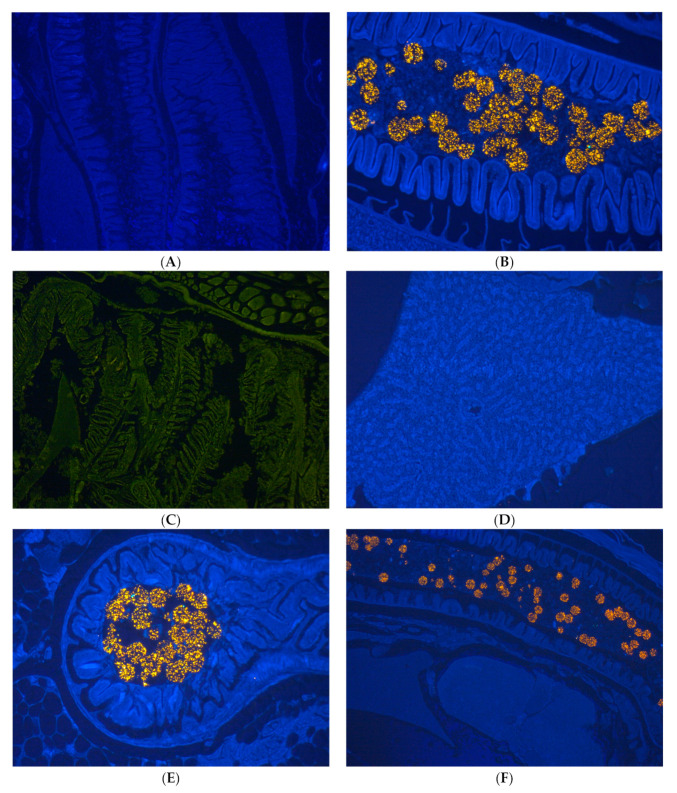
(**A**) Intestine of zebrafish, control. 40× Magnification. (**B**) Polyethylene MPs in the intestine of adult zebrafish exposed to 50 mg.L^−1^ for 96 h, 80× magnification. (**C**) MPs were not observed in gills of zebrafish exposed to 50 mg.L^−1^, 40× magnification. (**D**) Absence of polyethylene MPs in liver of zebrafish exposed 50 mg.L^−1^, 40× magnification. (**E**) Lumen of intestine of zebrafish full of MPs after exposure to 50 mg.L^−1^ for 96 h, 80× magnification. (**F**) Presence of MPs in the intestine; MPs were not ob-served in the other organs. 60× magnification.

**Figure 6 ijerph-20-03617-f006:**
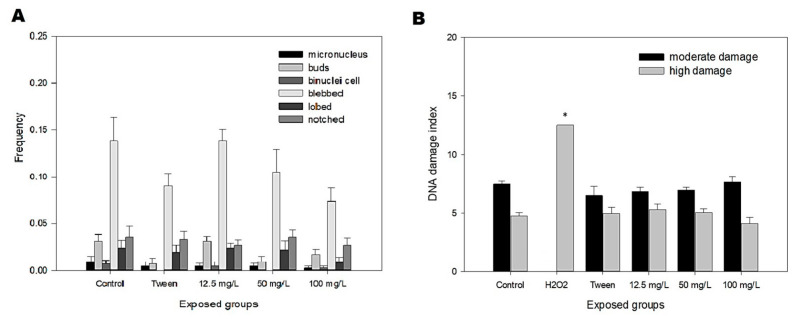
(**A**) Frequencies of micronucleus and nuclear abnormalities in peripheral erythrocytes of zebrafish adults. The result shows no statistical significance (*p* > 0.05). (**B**) Result of the comet assay. Moderate damage means tail length 1, 2 and high. H_2_O_2_ was highly positive (* *p* < 0.05).

**Figure 7 ijerph-20-03617-f007:**
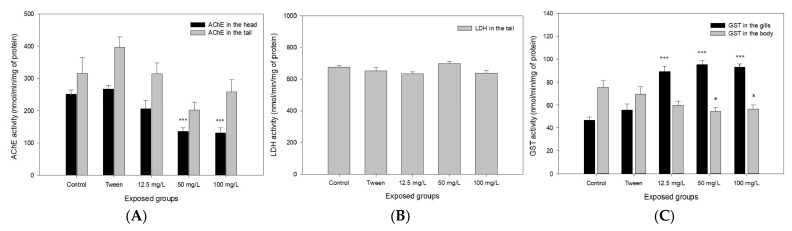
(**A**) AChE activity in the head and in the tail (mean values + standard error) of zebrafish adults after 96 h of exposure to polyethylene microplastics. Asterisks mean significantly different from the respective control (*p* < 0.001). (**B**) LDH activity in the tail (*p* > 0.05), and (**C**) GST activity in the gills (*** *p* < 0.001) and body (* *p* < 0.05).

## Data Availability

Not applicable.

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
