# Peer review of "Multilevel Toxicity Evaluations of Polyethylene Microplastics in Zebrafish (Danio rerio)"

_ijerph, 2023, doi:10.3390/ijerph20043617_

Round 1

Reviewer 1 Report

This study investigated the toxicity, genotoxicity, interaction and behavior changes of polyethylene MP beads in zebrafish.

These results present scientific evidence that the oxidative stress caused by a possible intestinal dysbiosis, which was not effective enough to induce genotoxicity and cytotoxicity.

This manuscript has academic significance as it extensively evaluated the effects of PE-MP spheres in adult, juvenile and embryo zebrafish.

Reviewer 2 Report

Manuscript Number: ijerph-2205225

Title: Multilevel toxicity evaluations of polyethylene microplastics in zebrafish (Danio rerio)

This paper studied multilevel toxicity of polyethylene microplastics (PE-MP) in adult zebrafish, including genotoxicity, cytotoxicity, histology and biochemical markers. They found that even though MPs caused physiological disturbances in zebrafish after short-term exposures and high concentrations in the intestine, MPs can be eliminated by zebrafish. I have following suggestions to improve the manuscript for possible publication. Here are my detailed comments, see below.

1.     Line 95: Why these concentrations were used? Any environmental implications of these concentrations?

2.     Figure 2: No error bars for the plot? Better to change Y-axis to %.

3.     Figure 2: Microplastic showed different effects on embryos with different concentrations and exposure time, why? I’d like to see more explanations.

4.     Figure 6: What’s the difference between “moderate damage” and “high damage”? How to define them?

Reviewer 3 Report

This manuscript tried to investigate the toxic effect of Polyethylene microplastics (PE-MP) in the range of 0.0; 6.25; 12.5; 50.0; 100.0 mg.L-1 on Zebrafish.

Here are some questions and comments to improve the overall quality of this manuscript:

Line 26: GI Tract:  The GI is an abbreviation that was used for the first time in this manuscript. It is recommended to use the complete word and add the GI in the bracket.

Line 76: “Fluorescent orange PE-MPs, diameter of 53 - 73 μm in average”. Why did the authors decide to move forward with this range of MPs? Why you didn’t select the smaller or larger size?

Line 83: Magnification of 200X while in Line 88 the magnification is 100X

Lines 90-93: What about light intensity and duration? Can light impact the process and toxicity?

Line 96: the selected range for MPs was from 6.25-100 mg.L-1. Why this range was selected?

Units: All the units should be consistent in the whole manuscript. Please fix the followings:

1.      mg.L-1 or mg/L please select one also for µg/g (line 256)

2.      mg.L-1 please superscript -1

3.      ºC please remove the underline from the degree symbol

4.      H2O2 , please subscript 2 (H2O2)

Line 109: 72 horas do the authors mean 72 hours?

Line 160: 05 min please remove 0 for consistency

Lines 177 and 277: figure 2, figure 4; Figure should be capital

Lines 270-275: What is the authors' conclusion of these results?
